# An Overview on Molecular Characterization of Thymic Tumors: Old and New Targets for Clinical Advances

**DOI:** 10.3390/ph14040316

**Published:** 2021-04-01

**Authors:** Valentina Tateo, Lisa Manuzzi, Claudia Parisi, Andrea De Giglio, Davide Campana, Maria Abbondanza Pantaleo, Giuseppe Lamberti

**Affiliations:** 1Department of Experimental, Diagnostic and Specialty Medicine, Policlinico di Sant’Orsola University Hospital, Via P. Albertoni 15, 40138 Bologna, Italy; valentina.tateo@studio.unibo.it (V.T.); lisa.manuzzi@studio.unibo.it (L.M.); claudia.parisi4@studio.unibo.it (C.P.); davide.campana@unibo.it (D.C.); maria.pantaleo@unibo.it (M.A.P.); giuseppe.lamberti8@unibo.it (G.L.); 2Division of Medical Oncology, IRCCS Azienda Ospedaliero-Universitaria di Bologna, Via P. Albertoni 15, 40138 Bologna, Italy

**Keywords:** thymic epithelial tumors, thymoma, thymic carcinoma, thymic neuroendocrine tumors, targeted therapy, molecular, biology, genomic, biomarkers, next-generation sequencing

## Abstract

Thymic tumors are a group of rare mediastinal malignancies that include three different histological subtypes with completely different clinical behavior: the thymic carcinomas, the thymomas, and the rarest thymic neuroendocrine tumors. Nowadays, few therapeutic options are available for relapsed and refractory thymic tumors after a first-line platinum-based chemotherapy. In the last years, the deepening of knowledge on thymus’ biological characterization has opened possibilities for new treatment options. Several clinical trials have been conducted, the majority with disappointing results mainly due to inaccurate patient selection, but recently some encouraging results have been presented. In this review, we summarize the molecular alterations observed in thymic tumors, underlying the great biological differences among the different histology, and the promising targeted therapies for the future.

## 1. Introduction

Primary Thymic Epithelial Tumors (TETs) are rare mediastinal tumors arising from thymic epithelial cells, with a reported annual incidence ranging from 1.3 to 3.2 per million [1]. TETs represent a heterogeneous group of malignancies, differing for their histological appearance and their biological behavior. According to the World Health Organization (WHO) classification, based on their morphology and the lymphocyte–to-epithelial cell ratio, TETs are classified into thymomas (TMs) and thymic carcinomas (TCs). TMs can be further categorized into five major subtypes with increasingly worse prognosis (types A, AB, B1, B2, B3), while all TCs are categorized into type C. According to the WHO classification of 2015, recognized TC subtypes are squamous cell, basaloid, mucoepidermoid, lymphoepithelioma-like, sarcomatoid, clear cell, adenocarcinoma, nuclear protein in testis (NUT), and undifferentiated [2,3,4]. TCs display the most aggressive behavior with a 5-year overall survival rate of only 50% [5]. Neuroendocrine tumors arising in thymus, or thymic neuroendocrine tumors (tNETs), first described in 1972 [6], are a distinct entity that will be discussed separately.

Clinical management in TETs is mainly driven by disease stage, with the Masaoka-Koga staging system currently routinely adopted for its optimal correlation with the prognosis [7,8,9]. CT and RMN are currently utilized for the diagnosis and staging of TETs, but some evidence about the usefulness of ^18^F-FDG-PET for the best planning of the treatment is available, showing a correlation between histological grade and ^18^F-FDG uptake [10,11]. Moreover, a positive correlation between ^18^F-FDG uptake and glucose transporter 1 (GLUT1), hypoxia-inducible factor-1 α (HIF-1α), vascular endothelial growth factor (VEGF), microvessel density (MVD), and p53 immunohistochemical (IHC) expression was observed [12].

A radical surgical resection represents the best treatment strategy for early-stage disease. On the contrary, the treatment of locally advanced or oligometastatic disease can be challenging and requires a multidisciplinary approach, including surgery, chemotherapy, and radiation therapy [13]. Systemic chemotherapy is the primary treatment for recurrent or metastatic disease. In TCs, the highest response rates are reported with carboplatin and paclitaxel [14], while the association of cisplatin, doxorubicin, and cyclophosphamide (CAP) is the preferred regimen for TMs [15]. Unfortunately, no standard salvage treatments are established for platinum-refractory patients. In addition, TETs represent a clinical challenge for the high rate of immune-mediated paraneoplastic syndromes, which also questions the use of immunotherapy approaches in these tumors [16].

In the last decade, the wide implementation of high throughput technologies in solid tumors has allowed the identification of a broad spectrum of molecular aberrations and altered signaling pathways in TETs, leading to the definition of distinct molecular profiles in TMs and TCs. The identification of specific aberrations in TETs has paved the way for novel therapeutic targeted strategies investigated in phase 1/2 studies [17]. However, the enrollment in trials of targeted therapies of patients with TMs together with those with TCs, not accounting for their distinct molecular profiles, prevented a conclusive interpretation of the results. Although no impressive changes in patients’ therapeutic paradigm progressed to platinum-based chemotherapy have been achieved so far, further efforts have been made to translate preclinical evidence into therapeutic targets. Hence, many novel trials are ongoing to implement precision medicine in the real world of TETs treatment [14].

In this review, we will summarize the genomic background of thymic tumors and the emerging molecular classification, with a significant focus on the biologic rationale explaining the possible use of targeted agents in this heterogeneous group of rare thoracic cancers (Figure 1). We will then focus on the ongoing studies and potential future perspectives based on previous studies’ results.

## 2. Overview of TETs Biology

The different histological subtypes of TETs harbor specific molecular alterations, as revealed by the comprehensive genomic analysis performed within The Cancer Genome Atlas (TCGA) project [18]. Indeed, TCs and TMs exhibit distinct molecular profiles and major oncogenic pathways involved in their pathogenesis. 

The genomic mutational profile of TETs is characterized by enrichment of C > T mutations within CpG islands, a mutational signature associated with aging and in agreement with the median age of onset [18]. 

Whole-Exome Sequencing (WES) on 117 samples of TETs and paired normal tissue has identified four recurrently mutated genes: *general transcription factor II-I* (*GTF2I*), *HRAS*, *TP53*, and *NRAS*. Clonality analyses revealed that the mutations in all four genes probably occurred at the onset or in the very early stages of tumor development. None of the four most frequently mutated genes in TETs is amenable for targeted inhibition to date [18]. 

Overall, TCs have been found to carry a higher number of mutations than TMs with recurrent mutations of known cancer-related genes, including *TP53*, *CYLD*, *cyclin-dependent kinase inhibitor 2A* (*CDKN2A*), *BRCA1 associated protein 1* (*BAP1*), and *polybromo 1* (*PBRM1*) [19]. Sequencing analysis of 409 genes performed in 12 different samples of TCs identified mutations in 24 genes, including *KIT*, *discoidin domain receptor tyrosine kinase 2* (*DDR2*), *platelet-derived growth factor receptor alpha* (*PDGFRA*), *ROS1*, *insulin-like growth factor 1 receptor* (*IGF1R*) [20]. DNA sequencing by targeted NGS of 174 patients with metastatic TC enabled the identification of clinically relevant genomic alterations specific for each sub-histology (squamous, non-neuroendocrine undifferentiated, neuroendocrine, adenocarcinoma, basaloid, lymphoepitheliomatous, and sarcomatoid carcinoma). Squamous, undifferentiated, and sarcomatoid subtypes harbored the highest number of genomic alterations, ranging between 4.1 and 4.8 on average; an average of 1.0 clinically relevant genomic alterations was then found, being *KIT* and *phosphatidylinositol-4,5-bisphosphate 3-kinase catalytic subunit alpha* (*PIK3CA*) the most frequently altered genes. Other targets included *PDGFRA*, *fibroblast growth factor receptor 3* (*FGFR3*), *protein patched homolog 1* (*PTCH1*), *F-box and WD repeat domain containing 7* (*FBXW7*), *breast cancer type 2 susceptibility protein* (*BRCA2*), *isocitrate dehydrogenase 1* (*IDH1*), *human epidermal growth factor receptor 2* (*ERBB2*), and *ERBB3* [21].

Regarding the tumor mutational burden (TMB), TETs are characterized by the lowest TMB among adult cancers. Comparing the TMB between TCs and TMs, a significant increase of TMB in TC samples could be observed [18]. In the study by Ross et al., the highest TMB was described in adenocarcinoma subtype (14% of cases had TMB greater than 10 mutations per Mb) and squamous cell carcinoma subtype (9% had a TMB greater than 20 mutations per Mb) [21]. Microsatellite instability, inducing very high TMB, has been exceptionally described in TCs [18].

Chromosomal copy number alterations (CNAs) have been described with high frequency in TETs and are usually associated with B2 and B3 TMs and with TCs [18]. Chromosomal aberrations correlate with WHO histologic classification and prognosis. For example, loss of chromosome 16q is typical of TCs [18], and the loss of the 6p23, where the tumor suppressor gene *forkhead box C1* (*FOXC1*) is codified [17], is associated with a shorter time to progression [22]. Furthermore, clusters of genomic aberrations correlate with the presence of autoimmunity. A higher level of aneuploidy was observed among patients with TMs presenting myasthenia gravis (MG). Moreover, MG was correlated with overexpression of genes, such as *medium-sized neurofilament* (*NEFM*) and *ryanodine receptor type III* (*RYR3*), presenting a sequence similarity with autoimmune targets [18].

Epigenetic alterations, such as aberrant DNA methylations, have been frequently observed and they also correlate with histological type and clinical stage. Silencing of tumor suppressor genes as *MLH1* by promoter hypermethylation, *O-6-methylguanine-DNA methyltransferase* (*MGMT*) methylation and loss of its protein expression as well as methylation of the promoter region of *CDKN2* have been frequently reported in TETs [17]. Non-coding RNAs (ncRNAs) are also involved in transcriptional and post-transcriptional regulation and their altered expression plays a role in the pathogenesis of several tumors, including TETs. Ganci et al. identified 87 microRNAs (miRNAs) differentially expressed between different TETs types and healthy tissues. Up-regulation of miRNAs promoting oncogenesis, such as miR-21-5p, and down-regulation of oncosuppressor miRNAs, as miR-145-5p, were also observed in TETs [23].

c-KIT is often overexpressed in TCs (79–88%), whereas *KIT* mutations are found in less than 10% of cases, with a wide spectrum of mutations not always sensitive to KIT inhibitors [18,24]. A similar pattern of overexpression concerns the epidermal growth factor receptor EGFR (71% TMs, 53% TCs) and HER2 (6% TMs, 47% TCs) [25] with few sensitizing mutations in these genes observed in TETs [17]. Activation of the PI3K/AKT pathway plays a pivotal role in TM growth and it may sensitize TETs to the inhibition of one of the key component of this intracellular axis, the serine-threonine kinase mammalian target of rapamycin (mTOR), as well as other specific inhibitors of the pathway [26]. IGF-1R is a transmembrane receptor able to increase the thymic epithelial cell population and influence the development of thymocytes and chemokine expression in the thymus [27]. The expression of IGF-1R, detected by immunohistochemistry (IHC), is often observed in TETs, especially in patients with recurrent or advanced disease and aggressive histologic subtypes (43% TMs, 86% TCs) [25,28]. As for many other solid tumors, angiogenesis plays an important role also in TETs and the overexpression of molecules belonging to the vascular endothelial growth factor receptor (VEGFR) family has been described in these cancers. Patients with TC display higher serum concentrations of VEGF and b-FGF than patients with TM [29]. TETs can also express somatostatin receptors (SSTR), providing a rationale for somatostatin’s anti-proliferative effect as a therapeutic option [30,31]. Additionally, mesothelin is expressed with high frequency in TCs and it is potentially targetable [32]. Proteomic characterization of TETs led to the identification of several proteins in TM with progressively different expression levels from normal thymus and across TM subtypes [33].

According to genomic analyses, TETs were classified into distinct molecular subtypes, with a good correlation with histological classification and prognosis (Table 1). 

Radovich et al. described 4 clusters of TETs based on a multi-omic unbiased clustering, which integrated mutation, CNA, mRNA, and miRNA expression, DNA methylation, and protein expression data. Subtype 1 (B-like) is principally represented by type B TMs and is characterized by *GTF2I* and *RAS* wild-type tumors, frequently associated with MG. Subtype 2 is mainly composed of TCs and tumors typically present chromosome 16q loss. Subtype 3 (AB-like) includes essentially type AB TMs and tumors generally are *GTF2I* mutated and *RAS* wild-type. Finally, subtype 4 (A-like) contains a mix of type A and AB TMs and is characterized by *GTF2I* and *RAS* mutated tumors [18]. Other analyses were performed by Lee et al., which identified 4 TET groups based on a supervised hierarchical clustering which integrated mutation, mRNA expression, and CNA data: the *GTF2I* mutant group is enriched in type A and AB TMs; the T-cell signaling gene profile group is composed principally by type B1, B2, and AB TMs; the chromosomally stable group includes mainly type B2 TMs; the chromosomally unstable group is principally represented by TCs and type B2 and B3 TMs. Both chromosomally stable and unstable clusters are enriched in MG cases. Interestingly, the T-cell signaling gene profile subgroup is enriched for genes related to costimulatory and coinhibitory T-cell signaling, implying an abundance of PD1-expressing CD8+ T cells, that may respond to immunotherapy [34]. These works present just a partial overlap between the two molecular classifications, but both show how histological subtypes significantly correlate with classes of genomic aberrations and demonstrate that A/AB-type, B-type, and C-type tumors are distinct biological entities rather than a histological continuum of diseases [18,34].

Based on molecular characterization of TETs, many targeted therapy clinical trials have been led (Table 2).

## 3. Characterization of TM Biology

As previously reported, there are recurrent mutations in some genes in TM that also define distinct subgroups of TETs. Those comprise *GTF2I*, genes of the PI3K/AKT/mTOR pathway, genes of the RAS family, and others. Other genomic alterations shared with TC are discussed in the respective section.

### 3.1. GTF2I

*GTF2I* is a gene located in the long arm of chromosome 7 at position 11.23 (7q11.23) [35]. It encodes for a multifunctional transcription factor (TFII-I/BAP-135), a protein that binds specific DNA regions to promote transcription in response to a variety of signals [36,37]. Several stimuli, for example from T- and B-cell receptors or growth factors pathways, can activate TFII-I by induction of tyrosine phosphorylation and cytoplasm to nucleus translocation. Translocation of activated TFII-I in the nucleus promotes the transcription of specific genes, such as FOS, a proto-oncogene involved in cell cycle regulating cyclin D1 gene transcription [38,39]. TFII-I is also implied in the endoplasmic reticulum (ER) stress response regulation, a cell defense mechanism in response to stress conditions targeting the ER. Its activation determines cell arrest, death induction, or promotion of anti-apoptotic pathways [40].

Mutations of *GTF2I* and gene fusions have been observed, respectively, in TETs and soft tissue angiofibromas (*GTF2I/NCOA2* fusion), acute promyelocytic leukemias (*GTF2I/RARA* fusion), and pilocytic astrocytoma (*GTF2I/BRAF* fusion) [35,38].

The *GTF2I* mutation in TETs always occurs at the same codon and seems to be pathognomonic, clonal, and oncogenic. Indeed, the L424H mutation has been observed only in TETs, while other tumors rarely present a mutation in *GTF2I* and always in different codons. Moreover, clonality analyses showed that this mutation is clonal, suggesting a very early onset in tumor development [18]. The *GTF2I* L424H mutation has a high prevalence in TETs (39–43.4%), especially in type A (82–100%) and AB (70–100%) TMs, while it is less frequent in more aggressive subtypes [18,19,41]. Furthermore, *GTF2I* mutation prevalence is associated with disease stage, since it is more frequent in early (57%) than advanced (19%) stages. Notably, TM patients with *GTF2I* mutant tumors lived longer than those with *GTF2I* wild-type tumors (10-year overall survival rate: 96% vs 88%) [19]. It is plausible that this mutation confers an indolent behavior to tumors, while more aggressive histology and late stages are typically characterized by other mutations, as discussed later, associated with a worse prognosis. Of note, the majority of publications on TETs molecular characterization analyzed surgical samples, so they are not exactly representative of molecular profile in advanced, not resectable tumors. Only a few works considered advanced stages samples [21,42].

The typical *GTF2I* L424H mutation observed in TETs affects a specific amino acid sequence, which is a non-canonical destruction box, involved in TFII-I proteasomal degradation. In presence of DNA damage, TFII-I is ubiquitinated and degraded by the proteasome complex. The L424H missense mutation determines a RILLAKE-to-RILHAKE alteration in the destruction box sequence that hampers TFII-I recognition for degradation. As a result, TFII-I turnover is reduced with consequent upregulation of downstream pathways, such as those involved in cell proliferation, cell morphogenesis, receptor tyrosine kinase signaling, retinoic acid receptors, neuronal processes, and the WNT and SHH signaling pathways [19,35,37]. On the other hand, apoptosis, cell cycle, DNA damage response, hormone receptor signaling, breast hormone signaling, RAS/MAPK, RTK, and mTOR pathways were downregulated [18,19,35,38]. Together, these factors suggest an oncogenic role of *GTF2I* in TETs [18].

Considering the high prevalence of the mutation and the oncogenic role in TETs, *GTF2I* with its correlated pathways could be a potentially interesting targeted treatment. At the moment, no targeted therapies have been developed for patients carrying this mutation.

### 3.2. PI3K/AKT/mTOR

The PI3K/AKT/mTOR signaling pathway is essential for the regulation of many cellular processes, such as proliferation, survival, metabolism, and angiogenesis, and is deregulated in many cancer types [43].

The PI3K/AKT/mTOR pathway activation plays a crucial role in TETs growth and can thus be exploited by drugs targeting mTOR or other inhibitors acting on the same pathway [44,45]. Mutations at different pathway levels, such as *PI3K*, *AKT*, *TSC*, and *mTOR*, have been observed in both TMs and TCs [44,45,46]. Mutations affecting proteins in the pathway are rare taken singularly, but taken together genomic alterations in the PI3K/AKT/mTOR pathway are present in more than 5% of TETs according to the TCGA PanCancer Atlas.

A phase 2 single-arm study of everolimus, an mTOR inhibitor, enrolled patients with TMs (N = 32) and TCs (N = 18) after at least one previous platinum-based chemotherapy [26]. The study met its primary end-point with a disease control rate (DCR) of 88% (76% stable disease, SD; 10% partial response, PR; 2% complete response, CR). When evaluated by histology, DCR was 94% in TMs, with 3 PRs, and 78% in TCs, with 1 CR and 2 PRs. The median progression-free survival (mPFS) was 16.6 and 5.6 months for TMs and TCs, respectively, and the median overall survival (mOS) was not reached for TMs and 14.7 months for TCs, respectively. However, safety has been an issue in this trial as 14 patients had a serious drug-related adverse event (AE) and 3 patients with TM died of drug-related pneumonitis [26]. In the pursue of predictive factors to identify patients more likely to respond to everolimus and optimize patient selection, pathogenic mutations were assessed by next-generation sequencing (NGS) on tumor samples from a small cohort of 15 pretreated patients with TET receiving everolimus. Pathogenic mutations in genes including *TP53*, *kelch-like ECH-associated protein 1* (*KEAP1*) and *CDKN2A*, were observed in 27% of patients, without association with time to treatment failure (TTF) [46].

PI3K inhibitors have been investigated in preclinical studies and showed potential activity in TETs [44]. A single-arm phase 2 trial of buparlisib in relapsed or refractory TMs was stopped early because of high toxicity and low efficacy: the overall response rate (ORR) was 7.1%, while G3-G4 AEs were reported in 50% of patients (NCT02220855).

### 3.3. IGF1R

The insulin-like growth factor (IGF) pathway regulates several biological processes, such as metabolism and cell growth. IGF I and II bind IGF-R1, which is a heterotetrameric transmembrane glycoprotein with an intracellular tyrosine kinase domain. IGF-R1 is encoded by a gene located in the long arm of chromosome 15 at position 26.3 (15q26.3) and is expressed ubiquitously, including in immune cells. The binding of IGF to IGF-R1 is modulated by the IGF binding proteins (IGFBPs 1-6) and leads to the activation of IGF-and of two major pathways: the insulin receptor substrate (IRS)/PI3K/AKT/mTOR pathway, with mainly metabolic effects, and the SHC/RAS/MAPK pathway, with mainly mitogenic effects [47,48].

In the thymus, IGF-1 has been shown to increase the thymic epithelial cell population and affect thymocyte development and chemokine expression [49]. Increased IGF-R1 activity in cancer is associated with the promotion of proliferation, migration, invasion, treatment resistance, and worse prognosis [50].

All histological subtypes of TETs have some degree of IGF-1R expression, especially aggressive subtypes and those at advanced disease stage [48,51]. Furthermore, a loss of heterozygosity of IGF-2R was frequently observed in TETs and may induce a compensatory upregulation of IGF-R1 [48]. 

Cixutumumab, a monoclonal antibody that binds IGF-1R and promotes its degradation, has been studied in a phase 2 trial in 49 pre-treated patients with TET (12 TCs and 37 TMs) [27]. In the TM cohort, 14% of the patients achieved a PR for a DCR of 89%, while none of the patients with TC responded, but 42% were stable. In respect to safety, 24% of patients with TM developed an autoimmune condition during treatment, the most common being pure red-cell aplasia. Severe AEs were reported in 31% of patients, and 2 patients died during treatment (one patient for respiratory failure, one patient for myositis, respiratory failure, and an acute coronary event). The most frequent G3-4 AEs were hyperglycaemia (10%) and increased serum lipases (6%) [27]. The high toxicity of IGFR inhibitors halted the development of these drugs in many cancers.

### 3.4. RAS

RAS proteins are a family of kinases with intrinsic guanosine-triphosphatase activity that mediate and integrate signal transduction from a multitude of cellular signals to their principal effectors that are RAF kinases and the PI3K/AKT/mTOR pathway. In humans, there are 4 isoforms of RAS, encoded by 3 genes: a gene located in the short arm of chromosome 12 at position 12.1 (12p12.1) encodes for 2 different splice variants of KRAS (KRAS4a and KRAS4b); a gene located in the short arm of chromosome 1 at position 13.2 encodes for NRAS (1p13.2); a gene located in the short arm of chromosome 11 at position 15.5 encodes for HRAS (11p15.5) [52]. *RAS* is mutated in 10–30% of all human cancers [53]: *KRAS* is mutated in 85% cases, while *NRAS* (12%) and *HRAS* (3%) are less common. The majority of *RAS* mutations occur at codons 12, 13, or 61 and determines constitutive activation of RAS [52]. 

RAS proteins are frequently mutated also in TETs (7–18.5%), especially HRAS and NRAS [18,24,54]. Overall, *RAS* mutations are more frequent in TCs than in TMs (18.5% vs 10%) [54]. *HRAS* mutation is more frequent in A/AB TMs, while *NRAS* is more frequent in TCs [18,55]. *RAS* mutations in TETs usually occur at known gain-of-function codons (e.g., *KRAS* G12A, *KRAS* G12V, *HRAS* G13V, *HRAS* G13R, *NRAS* G12D) and are associated with worse prognosis [18,24,54]. Recently, an allele-specific covalent inhibitor of KRAS G12C (AMG 510) has been developed, showing promising results in non-small cell lung cancer [56,57,58]. Although this could hopefully pace the way to the inhibition of other RAS alleles in other tumor types, currently there are no trials of RAS inhibitors in TETs.

### 3.5. Other Targets

SSTRs are expressed in TETs, thus the efficacy of octreotide, a somatostatin analog, with and without prednisone has been investigated by three phase 2 studies [30,31,59]. The primary endpoint was the ORR in each study, and was 37%, 31.6%, and 88%, respectively. Notably, no responses were reported in TCs. According to these findings, octreotide could be considered a therapeutic option in TMs with SSTR expression at functional imaging [30,31,59].

Members of the SRC family are tyrosine kinases involved in the transduction of signals for embryonal development and cellular growth. In the thymus, SRC is involved in thymic epithelial cell maturation. SRC role in the development of many types of cancer is well established so that SRC inhibitors have been developed [60]. Saracatinib is a highly selective small molecule that inhibits SRC and ABL whose activity profile has been investigated in a phase 2 trial that enrolled 21 pretreated patients with TET (N = 9 TCs and N = 12 TMs) [61]. The trial was halted after the first stage of accrual because no objective response was achieved. SD was observed in 8 patients with TM and 1 patient with TC at the first 8-week evaluation [61].

EGFR has a central role in the regulation of epithelial tissue development and homeostasis and its deregulation is observed and successfully targeted in many cancer types [62,63,64]. TETs commonly show high expression of EGFR at IHC but *EGFR* mutations are rare [24,54,65]. Response to anti-EGFR targeted therapy, such as cetuximab and apatinib, has been reported in case reports [66,67,68], but no objective responses were observed in a phase 2 trial of erlotinib and bevacizumab in 18 patient with refractory TET (11 TMs, 7 TCs) [69].

## 4. Characterization of TC Biology

As extensively showed, TC has a different biologic and mutational landscape compared to TMs within TETs. The most relevant and characterizing alterations occur in *KIT*, *CYLD*, and angiogenesis-related genes, that are also offer targets for specific treatments. Moreover, TCs show alterations in tumor suppressor genes (e.g., *TP53* and *RB1*) and in epigenetic regulators. Other genomic alterations shared with TM are discussed in the respective section.

### 4.1. KIT

The proto-oncogene *KIT* is a gene located in the long arm of chromosome 4 at position 12 (4q12) and encodes for a type III receptor tyrosine kinase, c-KIT (CD117), involved in many cellular processes. Binding with its ligand, the stem cell factor (SCF), triggers c-KIT dimerization and autophosphorylation of the tyrosine residues, protein kinase activation, and downstream activation of many signal transduction pathways including MAPK, PI3K/AKT/mTOR, PLCγ/DAG/IP3, JAK/STAT, and SRC pathway, with consequent stimulation of cell survival, proliferation, motility/invasion and angiogenesis [70,71]. Mutations in c-KIT commonly occur within the membrane region near the dimerization domain, codified by exon 8 and exon 9, and the intracellular tyrosine kinase domain, codified by exon 17. These gain-of-functions are associated with the development of gastrointestinal stromal tumors (GIST), germ cell tumors, melanomas, mastocytomas, and some leukemias and lymphomas [72,73,74,75,76,77].

Among TETs, TCs frequently present c-KIT overexpression (46–79%), whereas *KIT* mutations are found in less than 10% of cases. On the other hand, c-KIT overexpression is rare in TMs (2–4%), and no known mutation, except for a *KIT* deletion in a patient with AB thymoma reported on TCGA PanCancer Atlas, have been reported [24,78,79,80]. Mutations of *KIT* reported in TCs that show different drug susceptibility are V560del at exon 11, H697Y at exon 14, L576P at exon 11, Y553N at exon 11, D820E at exon 17, V559G at exon 11, 577–579del at exon 11, and K642E at exon 13 [24,65,80,81,82,83,84]. This wide spectrum of mutations is not always sensitive to c-KIT inhibitors: V560del, V559G, Y553N, and L576P mutations (all at exon 11) confer sensitivity to imatinib [24,65,81,83,85]; H697Y mutation at exon 14 shows resistance to imatinib but sensitivity to sunitinib [24]; D820E mutation at exon 17 and K642E mutation at exon 13 confer resistance to imatinib but are sensitive to sorafenib [80,82]; lastly, TC with 577–579del in exon 11 are sensitive to sorafenib [84].

Two phase 2 trials investigated the activity of imatinib in pre-treated patients with TETs. Neither study met its primary endpoint (ORR), but patients were not selected by the presence of *KIT* mutation [86,87]. Indeed, objective responses were observed in single-case reports of patients harboring *KIT* mutations sensitive to imatinib [81,83,85]. Further trials including patients selected by the presence of *KIT* sensitive mutations should be designed in order to assess the real activity of imatinib. However, the overall rarity of TETs and *KIT* mutations in these tumors makes a prospective trial unfeasible.

Sunitinib, a multikinase inhibitor of VEGFR, c-KIT, and PDGFR among others, was studied in a phase 2 trial with promising results [29]. The study met its primary endpoint in the TC cohort with an ORR of 26% (SD in 65%), while ORR was only 6% in the TM cohort (SD in 75%). The mPFS was 7.2 months and mOS was not reached within the TC cohort, while mPFS was 8.5 months and mOS 15.5 months within the TM cohort. Additionally, sunitinib treatment determined an increase of expression of PD-1 on circulating regulatory T cells and of cytotoxic T-lymphocyte-associated protein 4 (CTLA-4) on circulating CD8+ T cells in most patients, which was associated with improved overall survival. However, the upregulation of immune checkpoint receptors, resulting from T-cell activation, may limit the T-cell antitumor immunity in TETs treated with sunitinib. Thus, a combination of sunitinib and immune checkpoint inhibitors may potentially enhance antitumor responses [29,88].

Sorafenib, another multikinase inhibitor of RAF, VEGFR, c-KIT, PDGFR, and other kinases [89], showed antitumor efficacy in case series of patients with refractory TCs, irrespective of the presence of *KIT* mutations [80,82,84,90]. A case series of 5 patients with metastatic pre-treated TC reported PR in 2 patients (40%), SD in 2 patients (40%), and PD in 1 patient (20%). The mPFS and mOS were 6.4 and 21.2 months, respectively. Of note, the tumor of only one of the two responding patients harbored a *KIT* mutation (D820E at exon 17) [90].

### 4.2. CYLD

The *cylindromatosis deubiquitinase* (*CYLD*) gene is located in the long arm of chromosome 16 at position 12.1 (16q12.1) [91]. It encodes for a deubiquitinating enzyme that removes Lys63-linked ubiquitin chains from ubiquitinated proteins [92]. Ubiquitination is a fundamental post-translational process that can direct protein degradation via proteasomal machinery, autophagy, intracellular protein trafficking, DNA damage responses, protein activation, or interaction between proteins via different types of ubiquitination [93]. CYLD predominantly modulates NFκ-B signaling so that CYLD loss-of-function results in constitutive activation of NFκ-B, with consequent overexpression of proinflammatory and prosurvival genes [91,93]. 

CYLD is also involved in thymus development, in particular in differentiation and maturation of thymic medullary epithelial cells (mTECs): in fact, CYLD is a positive regulator of T-cell receptor signaling during the double-positive to single-positive transition of thymocytes, and also controls the nuclear entry of Bcl-3. Moreover, CYLD regulates the AIRE (AutoImmune REgulator transcriptional factor) expression in mTECs that is crucial for T-cell development [94]. 

Because CYLD has a central role in inflammation, cell death, cell cycle progression, cell migration, DNA damage, and WNT signaling, CYLD loss-of-function is associated with the deregulation of NFκ-B, JNK, c-MYC, and AKT and consequent tumor development [93,95], such as melanoma, leukemias, and TETs [92,96,97].

*CYLD* mutation in TETs is more frequent than in other tumors, especially in TCs with a prevalence >10%. CYLD-deficient TET cells, in presence of INFγ, upregulate PD-L1 via AKT-mediated increased STAT1 expression and increased activity of the STAT-IRF1 axis. CYLD loss also determines an increase in IRF1 in an INFγ-independent way by an increase in the basal Lys63-linked ubiquitination and consequent AKT activation. Activated AKT phosphorylates GSK3β and prevents it from phosphorylating IRF1, leading to missed IRF1 ubiquitination. The final effect is an increase in IRF1 half-life and activity [92,98]. Overall, CYLD loss led to increased PD-L1 expression in TET cells through both these cascades, with a significant correlation between low IHC CYLD expression and high PD-L1 expression (tumor proportion score ≥ 50%) [92]. Of note, this translates into better response to immune checkpoint inhibitors (ICIs), as observed in a phase 2 trial of pembrolizumab [92,99]. *CYLD* mutation was identified in 5 of the 36 tumors in which targeted exome sequencing was conducted, and it was associated with high PD-L1 expression. A post-hoc analysis showed a non-significant trend between *CYLD* mutation and longer PFS and OS [99].

These findings suggest that *CYLD* mutation or loss might serve as a potential biomarker of response to ICIs to better patient selection.

### 4.3. Angiogenesis

Angiogenesis is one of the hallmark of cancer and is sustained by the production of several growth factors, such as VEGF, PDGF, transforming growth factor beta (TGFβ), and angiopoietins. Nevertheless, tumor vessels are usually immature and defective, with consequent induction of hypoxia, decreased immune cell infiltration, increased risk of tumor dissemination, and reduced efficacy of drugs and radiotherapy [100,101].

Remarkably, the expression of vascular growth factors and their receptor has been observed in TETs, with a correlation between high levels and aggressive histology types [102]. Recently, dysregulation in the Activine A/Follistatin axis has been reported in TETs. Activine A is a member of the TGFβ superfamily that activates SMAD proteins and gene transcription, while Follistatin antagonizes and degrades Activine A. By the inhibition of Activine A, Follistatin promotes cell proliferation, tumor growth, and angiogenesis. Patients with TETs have higher Activin A and Follistatin serum concentrations than healthy controls. Follistatin levels were highest in patients with TCs and advanced tumor stage, and significantly correlated with tumor MVD [103]. 

As angiogenesis has a central role in cancer development and progression, many drugs targeting this process have been developed and are currently available and approved for different cancer types [104,105]. Bevacizumab, a humanized monoclonal antibody against VEGF, was investigated in a phase 2 trial in combination with erlotinib that enrolled 18 patients with recurrent TM (N = 11) or TC (N = 7). No objective responses were observed, SD was observed in 11 patients (60%), while in 7 patients (40%) PD was the best response [69]. More interesting results have been observed with multikinase inhibitors. The other multi-kinase inhibitors, sunitinib and regorafenib, with an antiangiogenic effect which also target c-KIT have been already discussed in the previous “KIT” paragraph [29,90]. In addition, the final results of the REMORA phase 2 trial have been recently reported [106]. The trial enrolled 42 patients with unresectable or metastatic TC who received at least one platinum-based chemotherapy, to evaluate the activity of lenvatinib, an oral multi-kinase inhibitor that targets VEGFR, FGFR, c-KIT, and other kinases. The trial met its primary end-point with an ORR of 38%. Of the 42 patients, 16 (38%) patients obtained a PR and 24 (57%) a SD. The DCR was 95%, the mPFS was 9.3 months and the mOS was not reached. The most frequent AEs were coherent with the well-known toxicity profile of lenvatinib: hypertension was reported in 88% of patients, decreased platelet count in 52% of patients, diarrhea in 50% of patients, and palmar-plantar erythrodysesthesia syndrome in 69% of patients. Serious AEs were reported in 8 (19%) patients, including bowel perforation, left ventricular dysfunction, pneumonitis, electrocardiogram T wave abnormalities, anorexia, and upper abdominal pain. There were no treatment-related deaths [106]. Lenvatinib is, to date, the most promising therapeutic option for thymic carcinoma patients progressing to standard first-line therapy. 

In respect to new antiangiogenic drugs, a recent case report described the efficacy of anlotinib, a multi-target tyrosine kinase inhibitor (TKI) that targets VEGFR, FGFR, PDGFR, and c-KIT, in a patient with refractory TC who achieved a SD with a PFS of 23 months [107]. However, more data is needed to assess the clinical utility of anlotinib in patients with TET.

### 4.4. Epigenetic Regulatory Genes and ncRNAs

Epigenetic processes regulate the transcriptional status of genes, chromosomal domains or entire chromosomes through chromatin remodeling, histone modification, DNA methylation/demethylation and interaction with ncRNAs, inducing a phenotype change without modifying the underlying DNA sequence [108]. TETs, and especially TCs, show mutations in many genes involved in epigenetic processes, namely *BAP1* (8%), *SET domain containing 2* (*SETD2*) (6%), *additional sex combs like 1* (*ASXL1*) (4%), *DNA methyltransferase 3 alpha* (*DNMT3A*) (4%), t*en-eleven translocation 2* (*TET2*) (4%), *Wilms tumor 1* (*WT1*) (4%) and *SWI/SNF related, matrix associated, actin-dependent regulator of chromatin, subfamily A, member 4* (*SMARCA4*) (3%). The prevalence of these mutations is higher when considering only TCs, being 13% for *BAP1* mutations, 9% for *SETD2* mutations, 6% for *DNMT3A* mutations, and 4% each for *ASXL1*, *SMARCA4*, *TET2*, and *WT1* mutations, respectively [42].

Methylation of DNA is the most studied epigenetic mechanism and is obtained by the addition of a methyl group to a cytosine residue in the context of CpG dinucleotides. CpG dinucleotides are present diffusively in the whole genome but are aggregated in CpG-rich regions, referred to as CpG islands. Methylation is catalyzed by different DNA methyl-transferases (DNMTs): DNMT1, DNMT3A, and DNMT3B. Methylation of CpG islands impedes interaction of DNA regulating regions with transcription factors or promotes recruitment of inhibitory proteins with the final effect of transcription silencing. On the other hand, demethylation, controlled by TET methylcytosine dioxygenases (TET1-3), eventually promotes transcription and has a fundamental role during embryogenesis [109,110,111]. Global DNA hypomethylation is typically associated with cancer, with consequent overexpression of oncogenes and chromosome instability, while hypermethylation silences tumor suppressor genes and promotes carcinogenesis [109,112]. *MGMT* is a frequently aberrantly methylated gene in TETs. This occurred far more commonly in TCs than TMs (74% vs 29%), with a significant association with loss of gene expression [113]. Methylation of *MGMT* promoter is correlated with a higher sensitivity to alkylating agents in different cancer types, including gliomas, lymphomas, and pancreatic NETs [114,115,116], suggesting a potential role as a predictive factor also in TETs.

Histones (H3, H4, H2A, H2B, and H1) are basic proteins with positive charge that pack DNA into repeating nucleosomal units and condensing them into chromatin. Histones are subjected to post-translational modifications, such as acetylation, phosphorylation, methylation, ubiquitination, and ADP-ribosylation, that modify the DNA-histone and histone-histone interactions, and regulate transcription by modulating access to chromatin by DNA translation machinery [117]. An unbalance among enzymes involved in histone post-translational modifications affects gene expression, as observed in some cancers [118]. Belinostat is a histone deacetylase (HDAC) inhibitor that has been investigated in two clinical trials enrolling TET patients [119,120]. A phase 2 study explored the activity of belinostat in TETs patients with recurrent or refractory disease. The trial enrolled 41 patients (N = 25 TMs and N = 16 TCs) showing a modest antitumor activity, with 2 PR (both in patients with TM), 25 SD, and 13 PD as best response [119]. The phase 1/2 trial of belinostat, before and in combination with CAP chemotherapy in first-line or recurrent TETs, enrolled 26 patients (N = 12 TMs and N = 14 TCs). An objective response was achieved in 64% of patients with TM and 21% of patients with TC [120]. It is important to note that no molecular selection of patients has been performed in these trials and that many epigenetic machineries are deregulated in TETs, thus explaining the reported low activity level of belinostat.

ncRNAs, such as microRNAs (miRNAs), long non-coding RNAs (lncRNAs), and circular RNAs (circRNAs), have a fundamental role in transcriptional regulation: miRNAs bind and inhibit mRNAs; lncRNAs interact with transcriptional regulation proteins, influencing chromatin structure and regulating mRNAs expression; circRNAs regulate mRNA splicing; all those ncRNAs have also a “sponge effect”, that allow a reciprocal regulation [121]. In cancers, ncRNAs have been identified as oncogenic drivers and tumor suppressors, since their dysregulation alters genetic expression [122]. A different expression of 87 miRNAs has been observed between TETs and normal thymus, but also between different histology subtypes. The upregulation of miR-21-5p and the downregulation of miR-145-5p have known pro-oncogenic activity as miR-21-5p targets the tumor suppressor *PTEN*, while miR-145-5p is a negative regulator of EGFR expression [23]. Deregulation of miRNAs in TETs is induced by epigenetic modifications. In fact, the use of HDAC inhibitors can enhance the expression of miR-145-5p, with consequent changes in expression levels of the pathway controlled by this miRNA [123]. The therapeutic role of miRNAs expression control should be further investigated in TETs.

*BAP1* is the most frequently mutated epigenetic regulatory gene in TCs. This tumor suppressor gene is located in the short arm of chromosome 3 at position 21.1 (3p21.1) and encodes for a deubiquitinating enzyme. Germline heterozygous *BAP1* mutations are responsible for the BAP1-cancer syndrome, an autosomal dominant condition characterized by high susceptibility to developing cancers, particularly uveal melanoma, malignant mesothelioma, cutaneous melanomas, renal cell carcinoma, and cholangiocarcinoma [124]. BAP1 deubiquitinase affects many cellular pathways, such as chromatin remodeling through ASXL1/2 interaction, with consequent histone H2A deubiquitination and repression of gene transcription, or DNA damage response with the deubiquitination of BARD1, which interacts with BRCA1 and regulates DNA repair. In mice models, BAP1 deletion determines severe thymic atrophy, complete loss of the T-cell, and impairment in B-cell development in the bone marrow, suggesting that BAP1 regulates thymic development and T-cell proliferation [124,125]. Since BAP1 loss-of-function sensitizes cells to DNA repair defects, the use of PARP inhibitors could be considered, especially in combination with or sequentially to therapies inducing double-strand DNA break, such as platinum-based chemotherapy, as observed in other cancer types [124]. Moreover, mutations of *BAP1* and other genes encoding epigenetic regulators may sensitize tumor cells to histone methyltransferase enhancer of zeste homolog 2 (EZH2) inhibitors, such as tazemetostat, which are currently entering early phases clinical trials but have not yet been investigated in TETs [126].

### 4.5. TP53

*TP53* gene is located in the short arm of chromosome 17 at position 13.1 (17p13.1) and encodes for the tumor-suppressor protein p53 [127], which controls the transcriptional regulation of genes involved in cell cycle arrest, apoptosis, senescence, DNA repair, and differentiation, but also in many other crucial cellular processes [128]. *TP53* is the most frequent mutated gene in human cancer since about 50% of tumors harbor a mutation in this gene. Mutations of *TP53* determine a loss-of-function in the onco-suppressive activity of the protein, but also a gain-of-function in oncogenic properties of the mutant p53 [128,129].

In healthy thymic epithelial cells, p53 is a key regulator of mTEC differentiation, through the RANK-NFκ-B pathway, and controls the expression of the tissue-restricted antigens. A p53 deficiency determines an aberrant thymopoiesis and an altered T-cell peripheral homeostasis with consequent abnormal immunological tolerance [130].

*TP53* is one of the most frequently mutated genes also in TETs, especially in TCs, in which mutations in *TP53* were reported in 18.5–26% of cases and are associated with worse outcomes with respect to *TP53* wild-type tumors [18,42,131,132,133].

Mutations in *TP53* are also associated with resistance to chemotherapy because of the involvement of multidrug resistance gene 1 (MDR1/ABCB1) [134]. Many molecules have been developed to target mutant p53, which can accelerate protein degradation or convert it into the wild-type conformation [129,134]. Drugs investigated in order to enhance the mutant p53 turnover are the heat shock protein 90 (HSP90) inhibitors, the HDAC inhibitors, and small molecules that target the mutant p53, inducing lysosomal degradation. Other molecules that can rescue wild-type p53 activity by promoting the proper folding of mutant p53 to restore the sequence-specific DNA binding capability, such as cysteine-binding compounds, Zn2+-chelating compounds, and specific peptides, are under investigation [129,134]. To date, no clinical trials however are currently ongoing in patients with TET.

### 4.6. CDK/RB

Cell-cycle transition through the four phases G1, S (DNA synthesis), G2, and M (mitosis) is strictly regulated by the cyclin-dependent kinases (CDKs) and upstream signaling pathways, such as mitogen, hormone, or growth factor stimulation. CDK3/cyclin C complex regulates the entry into cell cycle from G0 (quiescence). In the early phase G1, Cyclin D activates CDK4 and CDK6, which phosphorylate the tumor suppressor Retinoblastoma (RB) protein. Phosphorylated RB releases the E2F transcription factors, resulting in gene expression required for transition into the S phase. The CDK2/cyclin E complex completes the transition from G1 to S phase. The progression through phase S is controlled by the CDK2/cyclin A complex, while phase G2 is regulated by the complex CDK1/cyclin A, and CDK1/cyclin B complex completes the mitosis process. The cyclin kinase inhibitors (CKIs), such as p16, p21, and p27, negatively regulate the cell-cycle progression [135,136].

Dysregulations in CDKs, cyclins, and CKIs are frequent in human cancers, leading to abnormal cell proliferation. Cell-cycle aberrations have been also described in TETs, with alterations mainly in the CDK/RB pathway [55,137]. The alterations most frequently reported in TETs are CNAs of *CDKN2A/B*, hyper-methylation of their promoter, and loss of expression of p16, p21, and p27 [137,138]. Furthermore, deletions of *CDKN2A* (9p21) lead to p16 decrease and CDK4/6 hyper-activation and are associated with a worse prognosis in TCs [137].

Inhibitors of the CDKs have been studied in many human cancers, and are currently approved for hormone-sensitive breast cancer treatment [139,140].

The role of milciclib, an oral inhibitor of CDKs, tropomyosin receptor kinase A (TRKA), and SRC family kinases, was investigated in two phase 2 studies [141]. The CDKO-125A-006 study (NCT01011439) enrolled 72 patients with B3 TM (27,8%) or TC (72,2%), pre-treated with one chemotherapy line. The CDKO-125A-007 study (NCT01301391) enrolled 30 patients with B3 TM (56.7%) or TC (43.3%), pre-treated with multiple chemotherapy lines [141]. These two studies met their primary end-point with a 3 month-PFS of 44.4% and 54.2%, respectively. The mPFS and mOS were 6.83 and 24.18 months for the former study, whereas mPFS was 9.76 months, and OS was not reached for the latter study. In addition, DCR (75.9% vs. 83.3%) and ORR (3.7% vs. 4.2 %) were similar among these trials [141].

### 4.7. XPO1

*Exportin 1* (*XPO1*) gene is located in the short arm of chromosome 2 at position 15 (2p15) and encodes for XPO1, a nuclear exporter of proteins and RNAs [142]. The exchange of molecules between the cytoplasm and nucleus is mediated by the nuclear pore complex (NPC). Small molecules diffuse passively through the NPC, while large molecules need a shuttling protein, such as XPO1. XPO1 together with the RAN GTPase recognizes the nuclear export signals on its targets in the nucleus and binds RAN-GTP. Then, the complex passes through the NPC, and the hydrolysis of RAN-GTP to RAN-GDP causes the cargoes release. XPO1 is involved in the nuclear exportation of several molecules, such as tumor suppressor proteins (e.g. p53, FOXO3A, BRCA1/2, p27), but also oncoproteins (e.g. SNAIL, cyclins, YAP1, c-ABL) and RNAs (e.g., rRNAs, ncRNAs, mRNAs). Moreover, XPO1/RAN complex has an essential role in mitosis since it is fundamental for mitotic spindles assembly [143]. TMs and TCs show moderate to high nuclear expression of XPO1. Overexpression of XPO1 is associated with aggressive histology subtypes, advanced stage, and poor outcome [144]. 

Selinexor is a selective inhibitor of XPO1 that promotes its proteasomal degradation and consequently forces the nuclear localization and functional activation of tumor-suppressor proteins, prevents the oncoprotein mRNA translation, and causes cell cycle arrest and apoptosis in malignant hematologic and solid tumor cells [145,146,147]. Interestingly, the expression of the tumor-suppressor miR-145 is significantly lower in pancreatic ductal adenocarcinoma than in normal pancreatic ductal cells. Selinexor increases miR-145 expression with consequent downregulation of its target genes, such as *EGFR* and *MYC* [148]. 

In TET cells, selinexor determined nuclear accumulation of the tumor-suppressor proteins FOXO3a, p53, and p27. Additionally, selinexor determined cell-cycle arrest through the shuttling of many proteins that regulate cell-cycle progression and apoptosis due to the induction of the pro-apoptotic proteins BIM and BAX. Of note, GTF2I is another target of XPO1 [144]. 

A phase 1 trial evaluated the safety and efficacy of selinexor in 189 patients with advanced solid tumors. Remarkably, 4 TET patients were included, one had a PR, and three patients had SD, making selinexor a potential drug of interest for the future of TET treatment [149].

## 5. Thymic Neuroendocrine Tumors

Thymic neuroendocrine tumors (tNETs) are rare primary thymic neoplasms characterized by neuroendocrine differentiation, accounting for 2% of all neuroendocrine tumors, and about 5% of all thymic malignancies [150]. tNET are classified into well-differentiated neuroendocrine tumors (typical and atypical carcinoids, based on mitotic count and absence/presence of necrosis) and poorly-differentiated tumors, such as large cell neuroendocrine carcinoma (LCNEC) and small cell cancer (SCC), which are high-grade and aggressive cancers [3,4]. Consequently, 5-year survival is 50–70% for well-differentiated forms, down to nearly 0% for the poorly-differentiated ones [151]. The staging of tNETs has been historically based on the Masaoka-Koga system, similarly to TETs. Nowadays, the tumor nodes metastases (TNM) system by the American Joint Committee on Cancer is also widely used [9,152].

The 25% of tNETs arise in patients affected by multiple endocrine neoplasia type 1 (MEN1), a genetic disorder that predisposes to developing different kinds of neuroendocrine tumors [153,154]. Approximately 30% of tNETs are asymptomatic and incidentally discovered for an unrelated cause or within the surveillance of MEN1 mutation [155]. When present, symptoms vary according to the extent of the disease. tNETs usually present as a mass in the anterior mediastinal compartment and they can be aggressive neoplasms with a tendency to invade adjacent structures, and a locoregional lymph node involvement is present in up to 50% of cases at diagnosis [156]. Paraneoplastic syndromes, which are so common in patients with TM, are rare in tNETs as well as endocrine secretion syndromes (less than 5%), the most frequent being carcinoid or Cushing syndrome, especially in the setting of metastatic disease [157].

Because tNET are rare tumors, data to guide optimal treatment are limited and came from small retrospective trials and case series. Surgery is still the only curative-intent treatment and a complete resection represents the most significantly favorable prognostic factor for survival. Adjuvant radiation therapy (RT) plays a role in subtotally resected or locally advanced unresectable nonmetastatic disease. The evidence supporting the benefit of adjuvant RT is limited and, although it is associated with improved local control, there is no evidence of a survival benefit [151,158]. For poorly differentiated neuroendocrine carcinomas, even those that are completely resected, international guidelines suggest chemoradiotherapy with a platinum/etoposide-based regimen, rather than RT alone [14]. Surgery should always be considered for recurrent and/or metastatic settings, whenever the disease is potentially resectable. If surgery is not feasible, there are several systemic treatment options whose evidence is mostly based on retrospective studies on a limited number of patients [14,151,158]. In well-differentiated tNETs with a somatostatin-receptor-positive disease (by IHC or SSTR imaging), long-acting somatostatin analogs should probably be chosen as first-line treatment. Because mTOR is commonly deregulated in neuroendocrine tumors [159], everolimus is an alternative first-line treatment in patients with tNET following the results of the RADIANT-4 trial [160]. At PD, there are no data for selecting or sequencing treatments: the most used are peptide receptor radioligand therapy (PRRT) using a radiolabeled somatostatin analog or temozolomide-based chemotherapy [14,160,161,162,163,164]. Chemotherapy with platinum-based regimens, like cisplatin or carboplatin plus etoposide or oxaliplatin plus fluorouracil, is usually suggested in poorly-differentiated tNETs [14]. 

Even if the knowledge of the genetic variability of TMs and TCs has been deepened in recent years, still today, little is known about tNETs molecular characteristics. Sakane et al. have sequenced by NGS with a panel including 50 common cancer-related genes 54 patients with thymic neoplasia, including 48 TCs and 6 tNETs. The authors reported no significant differences in mutation frequency between TC and tNETs. The 3 most frequently mutated genes were *TP53* (18.5%), followed by *KIT* (7.4%) and *PDGFRA* (5.6%), which are commonly altered also in TETs [132]. Currently, there are no ongoing trials designed explicitly for tNETs. 

## 6. Future Perspectives

As there is no standard treatment for patients with advanced TETs after PD to platinum-based chemotherapy, several strategies, including targeted molecules, are being explored in different therapeutic settings (Table 3).

As surgery has a prominent role in the therapeutic strategy and outcome of thymic malignancies, neoadjuvant treatment with the aim of reducing tumor size and improving surgical outcomes might translate into better overall survival. With this aim, the preoperatory association of cetuximab and neoadjuvant chemotherapy (CAP regimen) in patients with resectable clinical Masaoka stage II-IVA TM or TC is under investigation in a phase 2 trial (NCT01025089).

In addition, many clinical trials are investigating the potential role of targeted therapies as single-agent therapies or in combination with other types of systemic treatment (e.g., immunotherapy). Indeed, the effectiveness of targeted agents in refractory/relapsed TETs, could be enhanced by adopting combination strategies, hence representing a promising approach.

### 6.1. TKI—Monotherapy

The multi-target TKI regorafenib has different targets involved in tumor angiogenesis and cell proliferation (e.g., VEGFRs 2 and 3, RET, c-KIT, PDGFR, and RAF kinases). A single-arm phase 2 trial (RESOUND) explores the activity of regorafenib in patients with different metastatic solid tumors refractory to available standard treatment, including TM (type B2–B3) and TC (NCT02307500).

Similarly, a phase 2 trial investigates the activity of sunitinib in patients with type B3 TM or TC who have received at least one prior platinum-containing chemotherapy regimen (Style Trial, NCT03449173).

### 6.2. TKI—Combination Therapy

To date, several trials are ongoing to better define the role of multi-targeted TKI in TETs when associated with other systemic treatments (e.g., chemotherapy, immunotherapy).

The RELEVENT study is an open-label phase 2 study of the combination of ramucirumab, carboplatin, and paclitaxel that will evaluate activity and safety in the first-line setting for relapsed or metastatic TETs of any histological type (NCT03921671). Of note, this study will evaluate the mutational status of a subset of genes, polymorphisms, and selected miRNA expression [165].

A phase 2 trial is assessing the activity of pembrolizumab, an anti-PD 1 monoclonal antibody, and sunitinib in participants with TC, not amenable to curative treatment (NCT03463460). Similarly, a multicentric, open-label, single-arm phase 2 study (PECATI study) will evaluate the efficacy and safety of the pembrolizumab-lenvatinib combination in pretreated immunotherapy-naïve patients with TC (NCT04710628).

Another phase 1/2 study will evaluate the safety and preliminary activity of the oral VEGFR/PDGFR kinase inhibitor vorolanib (CM082) combined with the anti-PD-1 nivolumab in patients with thoracic malignancies, including TC (NCT03583086).

In conclusion, the combination of TKI and immunotherapy may represent a promising strategy, although TKI induced cardiotoxicity risk could overlap with immune-mediated cardiological toxicity risk [16]. The toxicity profile is however a possible limit of this combination strategy in clinical practice, especially considering that the majority of pretreated patients will have received anthracycline-containing regimens.

### 6.3. Promising Monotherapy Other Than TKI

Nucleocytoplasmic transport is often altered in TETs [144]. An ongoing phase 2 trial will evaluate the activity of selinexor in patients with advanced TETs after PD to at least one platinum-containing chemotherapy regimen (NCT03193437).

Deregulation of TGF-β signaling pathway is observed across tumor types, as a consequence of increased expression of TGF- β or mutations/deletions of other axis components (TβRII, TβRI, Smad2, Smad3, Smad4) [166]. Multiple TGF-β pathway antagonists are at different preclinical and clinical development stages, with limited success so far. The bifunctional antibody bintrafusp alfa (M7824) consists of a PD-L1 region fused via a peptide linker to the TGF-β trap composed of the extracellular domain of TβRII, thus simultaneously binding both PD-L1 and TGF-β. Preclinical studies have shown bintrafusp alfa can enhance antitumor activity alone and combined with radiation, chemotherapy, and other immunotherapy agents [167]. Intravenous bintrafusp alfa is tested in a phase 2 trial, including patients with TET progressing after platinum-based chemotherapy (NCT04417660).

Mesothelin is a tumor differentiation antigen frequently overexpressed in tumors such as mesothelioma, ovarian, pancreatic, and lung adenocarcinomas. Anetumab ravtansine (BAY 94-9343) is an antibody-drug conjugate directed against mesothelin expressing cancer cells and able to induce a bystander effect on neighboring mesothelin-negative tumor cells, displaying encouraging preliminary antitumor activity in heavily pretreated patients [168]. A Phase 1b basket Study (ARCS-Multi) is investigating anetumab ravtansine among patients affected by advanced or recurrent malignancies, including mesothelin-expressing TC (NCT03102320).

## 7. Conclusions

Thymic neoplasia are rare malignancies with limited therapeutic options. Recent advances in the understanding of TET biology fostered by the wide access to new technologies, such as NGS, allowed to identify features underpinning molecular differences between TM and TC, some of which also served as potential targets for specific treatments. Despite the dramatic clinical and biological diversity, many clinical trials had enrolled patients with either histology due to the rarity of TETs. A short-term goal for TETs investigation should be tailoring clinical trials to histology and molecular subtypes. Recently, promising results from targeted therapies and immunotherapy have been reported, but safety and appropriate response biomarkers identification are still open questions. Although several studies have been led and targets have been identified, no targeted treatment is currently approved for TET patients in Europe.

## Figures and Tables

**Figure 1 pharmaceuticals-14-00316-f001:**
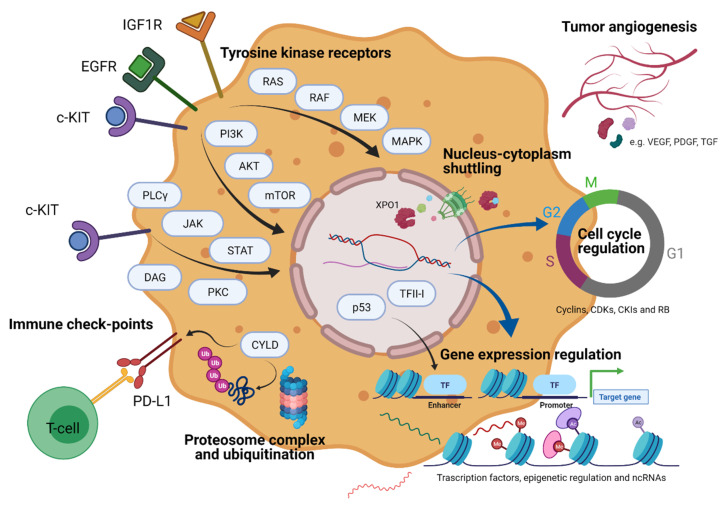
Main molecular pathways involved in the pathogenesis of thymic epithelial tumors. Created with BioRender.com.

**Table 1 pharmaceuticals-14-00316-t001:** Principal molecular classifications in Thymic Epithelial Tumors (TETs).

First Author, Year	Molecular Subtypes	Typical Genomic Profile	Enriched for MG	Main Histotypes	Prognosis
Radovich M et al. 2018	1 (B-like)	wtGTF2I, wtRAS, ↓p53, ↑MYC/MAX, ↓PPARA-RXRA, ↓XBP1-2, ↑MYB	+	B	Intermediate
	2 (C-like)	wtGTF2I, wtRAS, chr16q loss, ↓p53, ↑MYC/MAX, ↓XBP1-2, ↓PPARA-RXRA, ↑MYB	₋	C	Poor
	3 (AB-like)	mGTF2I, wtRAS, ↑C19MC, ↑MYB, ↓p53, ↑FOXM1, ↓TAp73a, ↑E2F1/DP	₋	AB	Good
	4 (A-like)	mGTF2I, mRAS, ↑C19MC, ↑p53, ↑XBP1-2, ↓MYC/MAX, ↓MYB, ↓FOXM1	₋	A and AB	Good
Lee HS et al. 2017	GTF2I	mGTF2I	₋	A and AB	Good
	TS	wtGTF2I, ↑genes associated with TS	±	AB, B1 and B2	Good
	CS	wtGTF2I, sCNA low	+	B2	Poor
	CIN	wtGTF2I, sCNA high, delCDKN2A	+	B2, B3 and C	Poor

Abbreviations: MG, myasthenia gravis; m, mutated; wt, wild-type; ↑, overexpressed; ↓, underexpressed; chr, chromosome; TS, T-cell signaling; CS, chromosomal stability; CIN, chromosomal instability; sCNA, somatic copy number alterations; del, deletion.

**Table 2 pharmaceuticals-14-00316-t002:** Published clinical trials of targeted therapy in TETs.

First Author, Year (Study Name)	Phase	TCs (n)	TMs (n)	Experimental Drug	mPFS	ORR, %	DCR, %	G3-G4 AEs n (%)
NCT02220855	II	0	14	buparlisib	11.1 months	7.1%	50%	7 (50%)
Rajan A et al. 2014 (NCT00965250)	II	12	37	cixutumumab	9.9 for TMs and 1.7 for TCs	14% for TMs and 0% for TCs	89% for TMs and 42% for TCs	29 (59.2%)
Palmieri G et al. 2002	II	6	10	octreotide and prednisone	14 months	37%	75%	0 (0%)
Loehrer PJ Sr et al. 2004 (NCT00003283)	II	6	32	octreotide ± prednisone	8.8 months for TMs and 4.5 months for TCs	37.5% for TMS and 0% for TCs	67.1%	8 (21.5% G4-5)
Kirzinger L et al. 2016 (NCT00332969)	II	2	15	octreotide LAR and prednisone	N/A	100% for TMs 0% for TCs	N/A	3 (17.6%)
Gubens MA et al. 2015 (NCT00718809)	II	9	12	saracatinib	5.7 months for TMs and 3.6 months for TCs	0%	42,9%	3 (14.3%)
Giaccone G et al. 2009	II	5	2	imatinib	2 months	0%	100% for TMs and 0% for TCs	2 (28.6%)
Palmieri G et al. 2012	II	3	12	imatinib	3 months	0%	8.3% for TMs and 0% for TCs	0 (0%)
Thomas A et al. 2015 (NCT01621568)	II	24	16	sunitinib	7.2 months for TCs and 8.5 months for TMs	26% for TCs and 6% for TMs	91% for TCs and 81% for TMs	28 (70%)
Bedano PM et al. 2008 (NCT00369889)	II	7	11	erlotinib and bevacizumab	N/A	0%	60%	N/A
Sato J et al. 2020 (REMORA trial)	II	42	0	lenvatinib	9.3 months	38%	95%	8 (19%)
Giaccone G et al. 2011 (NCT00589290)	II	16	25	belinostat	5.8 months	8% for TMs and 0% for TCs	25%	6 (14.6%)
Thomas A et al. 2014 (NCT01100944)	I/II	14	12	belinostat and chemotherapy	not reached for TMs and 7.2 months for TCs	64% for TMs and 21% for TCs	100% for TMs and 93% for TCs	20 (76.9%)
Besse B et al. 2018 (NCT01011439)	II	52	20	milciclib	6.8 months	3.7%	75.9%	22 (30.6%)
Besse B et al. 2018 (NCT01301391)	II	13	17	milciclib	9.8 months	4.2%	83.3%	14 (46.7%)
Abdul Razak AR et al. 2016 (NCT01607905)	I	0	4	selinexor	N/A	25%	100%	N/A

Abbreviations: TCs, thymic carcinomas; TMs, thymomas; mPFS, median progression free survival; ORR, overall response rate; DCR, disease control rate; AEs, adverse events; N/A, data not available. Note: data presented as No (%).

**Table 3 pharmaceuticals-14-00316-t003:** Ongoing clinical trials of targeted therapy in TETs (source: clinicaltrials.gov; last accessed: 10 February 2021).

Trial	Phase	Disease	Setting	Experimental Arm	Estimated Enrollment	Primary Endpoint
NCT03102320 (ARCS-Multi)	Ib	Thoracic tumors including TC	Pre-treated	anetumab ravtansine	173	ORR
NCT03583086	I/II	Thoracic tumors including TC	Pre-treated	vorolanib + nivolumab	177	Safety, ORR
NCT01025089	II	Locally Advanced or Recurrent TC or TM	Neoadjuvant	cetuximab + CAP	18	cPR
NCT03921671 (RELEVENT Trial)	II	TC and B3 TM	Advanced, untreated	ramucirumab + carboplatin and paclitaxel	60	ORR
NCT02307500 (RESOUND Trial)	II	Solid Tumors including TC and B2-B3 TM	Pre-treated	regorafenib	82	2-months PFS rate
NCT03449173 (Style Trial)	II	TC and B3 TM	Pre-treated with Platinum-based CHT	sunitinib	56	ORR
NCT03463460	II	TC	Pre-treated with Platinum-based CHT	pembrolizumab + sunitinib	40	ORR
NCT04710628 (PECATI)	II	TC and B3 TM	Pre-treated with Platinum-based CHT	pembrolizumab + lenvatinib	43	mPFS
NCT03193437 (SELECT trial)	II	TC and TM	Pre-treated with Platinum-based CHT	selinexor	25	ORR
NCT04417660	II	TC and TM	Pre-treated with Platinum-based CHT	bintrafusp alfa	38	ORR

Abbreviations: TC, thymic carcinoma; TM, thymoma; mPFS, median progression free survival; ORR, overall response rate; CAP, cisplatin, doxorubicin, cyclophosphamide; cPR complete pathologic response. Note: data presented as No.

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
