# Peer review of "An Overview on Molecular Characterization of Thymic Tumors: Old and New Targets for Clinical Advances"

_pharmaceuticals, 2021, doi:10.3390/ph14040316_

Round 1

Reviewer 1 Report

The authors performed a comprehensive review that wants to summarize the genomic background of thymic tumors and the emerging molecular classification, with a significant focus on the biologic rationaleexplaining the possible use of targeted agents in this heterogeneous group of rare thoracic cancers. In particular their review is focused on the ongoing studies and potential future perspectives based on previous studies' results.

It’s a fine work and properly written paper that carries an interesting sound message and some additional clinical information about this still unknown neoplasm. 

General considerations:

This study has the great merit to analyze in deep the genomic background of thymic tumors and identified four recurrently mutated genes: general transcription factor II-I (GTF2I), HRAS, TP53 and NRAS. Interestingly, the mutations in all four genes seem to occur at the onset or in the very early stages of tumor development and especially in less aggressive subtype. As reported in the review, the GTF2I L424H mutation has a high prevalence in TETs (39-43.4%), especially in type A (82-100%) and AB (70-100%). Notably, TM patients with GTF2I mutant tumors lived longer than those with GTF2I wild-type tumors (10-year overall survival rate: 96% vs 88%) [15].

This is a crucial point because, from a clinical point of view, TETs benefit from surgery in the most part of cases excluded locally-advanced/metastatic tumors. Unfortunately, this mutation seems to be not frequent in this scenario where a new target may be very useful.

Please, discuss in deep this topic in the review because future perspectives probably depends on it

.

Few minor revisions should be considered:

1: An other interesting topic that I suggest to briefly report in the review concern the correlation between histology and radiometabolic results. In detail, an accurate histological classification (and accordingly mutational profile) is not adequate on pre-operative biopsy but only on surgical specimen. Physicians usually adopt PET TAC with FDG to have a “non-surgical biopsy” of the thymic lesion before surgery. It could be interestingly review if mutational staus has been correlated with PET findings since this may potentially have a clinical impact of overall management of TETs.

We suggest to include to use this meta-analysis (Treglia G, Sadeghi R, Giovanella L, Cafarotti S, Filosso P, Lococo F. Is (18)F-FDG PET useful in predicting the WHO grade of malignancy in thymic epithelial tumors? A meta-analysis. Lung Cancer. 2014 Oct;86(1):5-13. doi: 10.1016/j.lungcan.2014.08.008. Epub 2014 Aug 18. PMID: 25175317.) as inspiration for this integration.

2: There is no mention of possible clusterization of genetic progile between TETs with Miasthenya gravis and TETs without it. Is there any evidence in literature? Please discuss on this topic also.

Author Response

  • General considerations: This study has the great merit to analyze in deep the genomic background of thymic tumors and identified four recurrently mutated genes: general transcription factor II-I (GTF2I), HRAS, TP53 and NRAS. Interestingly, the mutations in all four genes seem to occur at the onset or in the very early stages of tumor development and especially in less aggressive subtype. As reported in the review, the GTF2I L424H mutation has a high prevalence in TETs (39-43.4%), especially in type A (82-100%) and AB (70-100%). Notably, TM patients with GTF2I mutant tumors lived longer than those with GTF2I wild-type tumors (10-year overall survival rate: 96% vs 88%) [15]. 
    This is a crucial point because, from a clinical point of view, TETs benefit from surgery in the most part of cases excluded locally-advanced/metastatic tumors. Unfortunately, this mutation seems to be not frequent in this scenario where a new target may be very useful.
    Please, discuss in deep this topic in the review because future perspectives probably depends on it;
    We agree with the reviewer on the importance of new targets for advanced TETs. GTF2I is a mutation typically associated with low-grade histologies, early stages, and consequently good prognosis. It's plausible that this mutation confers an indolent behavior to tumors, while more aggressive histologies and late stages are typically characterized by other mutations, analyzed in the manuscript, associated with a worse prognosis. Of note, the majority of publications on TETs molecular characterization analyzed surgical samples, so they are not exactly representative of molecular profile in advanced, not resectable tumors. The only two works that analyzed the molecular profile in advanced TETs are by Wang 2014 and Ross 2017. We stressed, as suggested, these important concepts in the manuscript.

  • An other interesting topic that I suggest to briefly report in the review concern the correlation between histology and radiometabolic results. In detail, an accurate histological classification (and accordingly mutational profile) is not adequate on pre-operative biopsy but only on surgical specimen. Physicians usually adopt PET TAC with FDG to have a “non-surgical biopsy” of the thymic lesion before surgery. It could be interestingly review if mutational staus has been correlated with PET findings since this may potentially have a clinical impact of overall management of TETs. We suggest to include to use this meta-analysis (Treglia G, Sadeghi R, Giovanella L, Cafarotti S, Filosso P, Lococo F. Is (18)F-FDG PET useful in predicting the WHO grade of malignancy in thymic epithelial tumors? A meta-analysis. Lung Cancer. 2014 Oct;86(1):5-13. doi: 10.1016/j.lungcan.2014.08.008. Epub 2014 Aug 18. PMID: 25175317.) as inspiration for this integration.
    We are grateful to the reviewer for this suggestion. CT and RMN are currently utilized for the diagnosis and staging of TETs, but some evidences about the usefulness of 18F-FDG-PET for a best planning of the treatment are available, showing a correlation between histological grade and 18F-FDG uptake. Moreover, a positive correlation between 18F-FDG uptake and glucose transporter 1 (GLUT1), hypoxia-inducible factor-1 α (HIF-1α), vascular endothelial growth factor (VEGF), microvessel density (MVD) and p53 immunohistochemical (IHC) expression was observed. No other correlations between FDG uptake and molecular findings are available, as far as we know.

  • There is no mention of possible clusterization of genetic progile between TETs with Miasthenya gravis and TETs without it. Is there any evidence in literature? Please discuss on this topic also.
    The study performed by Radovich et al showed that genomic aberrations correlate with the presence of autoimmunity. A higher level of aneuploidy was observed among patients with TMs presenting myasthenia gravis (MG). Moreover, MG was correlated with overexpression of genes, such as NEFM and RYR3, presenting a sequence similarity with autoimmune targets. The analysis performed by Lee et al showed enrichment in MG+ cases in the chromosomally stable and instable clusters. We discussed this topic in paragraph 2. 

Reviewer 2 Report

The paper by Tateo V et al addresses molecular features and the published and ongoing clinical trials of targeted therapy in thymic epithelial tumors (TETs) - Tymomas (TMs); Thymic carcinomas (TCs); and Thymic neuroendocrine neoplasms: well-differentiated tumors (tNETs) and carcinomas (tLCNEC and tSCC) .

The manuscript (MS) might be of best interest for readers if the authors clarify/state some issues:

  • The “sub-histology” of TCs should follow the current WHO classification (e,g. neuroendocrine is considered an independent group, as stated in the manuscript);
  • Clarify/state in a separate Table the main molecular type classifications, according to the respective genetic alterations;
  • Clarify/state the main targeted therapies according to the molecular classification(s) as applicable;
  • Present a separate Table with the main best results (e.g., objective response rate-ORR %; disease control rate-DCR) and worst adverse effects-AE, by TETs subgroups;
  • Present/clarify the putative limitations (e.g., the inaccurate patient selection factors, as indicated in the abstract) that so far preclude the approval of targeted therapies for the next future.

Author Response

  • The “sub-histology” of TCs should follow the current WHO classification (e,g. neuroendocrine is considered an independent group, as stated in the manuscript);
    We are grateful for this suggestion. According to the WHO classification of 2014, recognized TC subhistologies are squamous cell, basaloid, mucoepidermoid, lymphoepithelioma-like, sarcomatoid, clear cell, adenocarcinoma, nuclear protein in testis (NUT), and undifferentiated. We inserted in paragraph 1 this clarification. 
  • Clarify/state in a separate Table the main molecular type classifications, according to the respective genetic alterations;
    We thank the reviewer for this comment. A table with the principal molecular classification was added (Table 1). 
  • Clarify/state the main targeted therapies according to the molecular classification(s) as applicable;  
    We thank the reviewer for this comment. As we stated in paragraph 2, Radovich et al. and Lee et al. described 4 clusters of TETs based on molecular classification. Due to the disease's rarity, trials testing targeted therapies have been conducted regardless of the molecular subtype. Aware of the pivotal role of the molecular characterizations, as you suggested, we structured the review discussing each molecular alteration and its clinical implication per time.
  • Present a separate Table with the main best results (e.g., objective response rate-ORR %; disease control rate-DCR) and worst adverse effects-AE, by TETs subgroups;
    We are grateful for this suggestion. Historically, clinical trials on TETs have never preplanned studies to explore the efficacy of drugs among molecular subtypes. Predominantly, inclusion criteria allowed the recruitment of TETs patients regardless of the histology. We created a comprehensive table in this scenario, including published trials testing targeted therapies and reporting endpoint results according to histology (Table 2).
  • Present/clarify the putative limitations (e.g., the inaccurate patient selection factors, as indicated in the abstract) that so far preclude the approval of targeted therapies for the next future.
    We agree with the reviewer that this is a point of paramount importance. In the conclusion section, we stated as follows: 'Despite the dramatic clinical and biological diversity, many clinical trials had enrolled patients with either histology due to the rarity of TETs. A short-term goal for TETs investigation should be tailoring clinical trials to histology and molecular subtypes. Recently, promising results from targeted therapies and immunotherapy have been reported, but safety and appropriate response biomarkers identification are still open questions.'

Reviewer 3 Report

This manuscript is a review of molecular characterization of thymic tumors. Biology of thymic epithelial tumors, biology of thymoma and thymic cancer, and biology of thymic neuroendocrine tumors are described concisely. This work is interesting to many readers of this journal.

I have no hesitation in recommending this manuscript for publication.

Author Response

The authors are grateful for your positive comments on the submitted manuscript. 

Round 2

Reviewer 2 Report

The MS in the revised form is of interest to readers.